# Obtaining Nanostructured ZnO onto Si Coatings for Optoelectronic Applications via Eco-Friendly Chemical Preparation Routes

**DOI:** 10.3390/nano11102490

**Published:** 2021-09-24

**Authors:** Mirela Petruta Suchea, Evangelia Petromichelaki, Cosmin Romanitan, Maria Androulidaki, Alexandra Manousaki, Zacharias Viskadourakis, Rabia Ikram, Petronela Pascariu, George Kenanakis

**Affiliations:** 1Center of Materials Technology and Photonics, Hellenic Mediterranean University, 71410 Heraklion, Greece; 2National Institute for Research and Development in Microtechnologies (IMT-Bucharest), 023573 Bucharest, Romania; cosmin.romanitan@imt.ro; 3Institute of Electronic Structure and Laser, Foundation for Research & Technology-Hellas, N. Plastira 100, 70013 Heraklion, Greece; epetromichelaki@physics.uoc.gr (E.P.); pyrhnas@physics.uoc.gr (M.A.); manousa@iesl.forth.gr (A.M.); zach@iesl.forth.gr (Z.V.); 4Physics Department, University of Crete, 71003 Heraklion, Greece; raab@um.edu.my; 5Department of Chemical Engineering, University of Malaya, Kuala Lumpur 50603, Malaysia; 6“Petru Poni” Institute of Macromolecular Chemistry, 700487 Iasi, Romania

**Keywords:** zinc oxide, green synthesis, nanostructured, optical properties, optoelectronics

## Abstract

Although the research on zinc oxide (ZnO) has a very long history and its applications are almost countless as the publications on this subject are extensive, this semiconductor is still full of resources and continues to offer very interesting results worth publishing or warrants further investigation. The recent years are marked by the development of novel green chemical synthesis routes for semiconductor fabrication in order to reduce the environmental impacts associated with synthesis on one hand and to inhibit/suppress the toxicity and hazards at the end of their lifecycle on the other hand. In this context, this study focused on the development of various kinds of nanostructured ZnO onto Si substrates via chemical route synthesis using both classic solvents and some usual non-toxic beverages to substitute the expensive high purity reagents acquired from specialized providers. To our knowledge, this represents the first systematic study involving common beverages as reagents in order to obtain ZnO coatings onto Si for optoelectronic applications by the Aqueous Chemical Growth (ACG) technique. Moreover, the present study offers comparative information on obtaining nanostructured ZnO coatings with a large variety of bulk and surface morphologies consisting of crystalline nanostructures. It was revealed from X-ray diffraction analysis via Williamson–Hall plots that the resulting wurtzite ZnO has a large crystallite size and small lattice strain. These morphological features resulted in good optical properties, as proved by photoluminescence (PL) measurements even at room temperature (295 K). Good optical properties could be ascribed to complex surface structuring and large surface-to-volume ratios.

## 1. Introduction

In optoelectronics, ZnO is commonly used as a polycrystalline nanostructured coating, thin film, or single crystals. As the optical character of the material depends on its structure, size, and shape of building blocks and the resulting surface structure that is a direct consequence of the preparation process, the main goal of this work is to compare the optical properties of various kinds of nanostructured ZnO coatings resulting from different kinds of syntheses, including also a few green routes. The comparison was done by evaluating PL spectral properties of ZnO layers grown onto Si substrates and relating them to the specific bulk and surface structure via respective synthesis routes.

The wurtzite-type structure of ZnO is the most thermodynamically stable form of this semiconductor and exhibits a wide bandgap at around 3.37 eV [1,2]. Compared to gallium nitride, which is a semiconductor with a similar energy gap, ZnO possesses strong polar binding, large exciton binding energy (60 meV), and relatively high longitudinal optical (LO) phonon energy of 72 meV [2]. Due to the presence of oxygen and zinc atoms interlaced along the c-axis, wurtzite-type ZnO is polar in this direction, thus gaining piezoelectric properties. Although the as grown ZnO exhibits n-type conductivity [1,2], the development of methods of n-doped and p-doped ZnO thin film deposition is still of vivid interest. Many deposition techniques were used during ZnO thin films history: physical growth methods, chemical methods, and combinations of them. Among chemical techniques, aqueous chemical growth, sol-gel, dip coating, and biosynthesis are the most common ones [3,4,5,6,7,8,9,10,11,12,13,14]. Due to the aforementioned properties, the ZnO and its composite materials are considered to be great candidates for light-emitting and laser devices, which cover the spectral range from green over the blue to the near-UV. In addition, sensors, optical filters, and nonlinear optics devices could be obtained by virtue of the remarkable ZnO properties [2]. Undoped ZnO layers exhibit high electron concentration and are quite insensitive to visible light; thus, they are also suitable as a semiconductor in thin-film transistors [2] or as transparent conductive electrodes for various applications [2]. Compared to ITO films, ZnO layers are chemically stable, can be safely fabricated at low-cost, and are harmless for living creatures [2]. Piezoelectric and ferroelectric properties of ZnO films allow them be used also in acoustoelectronic, acousto-optic, and memory devices [1,2]. Although the research on ZnO has a very long history and its applications are almost countless as the publications on this subject are extensive, this semiconductor is still full of resources and continues to offer very interesting results worth publishing or warrants further investigation [15,16]. The recent years are marked by the development of novel green chemical synthesis routes for semiconductors fabrication in order to reduce the environmental impact of fabrication on one hand and reduce the toxicity and hazards at the end of their lifecycle on the other. Green chemical syntheses are targeting the design of chemical products and processes that reduce or eliminate the use or generation of hazardous substances. Green chemistry applies across the life cycle of a chemical product, including its design, manufacture, use, and ultimate disposal. Reducing or eliminating one or more high purity solvents or reagents from a process result in a whole chain of substantial energy consumption, reduced use of hazardous and toxic materials consumption used in the synthesis and purification of the respective solvent or reagent, and generates long term positive effects on reducing the environmental impact of chemical processing. As an example, ethanol is one of the most used and cheap solvents, and it is synthesized by the hydration of ethylene and by fermentation. In semiconductor fabrication, the highest purity of materials is required; thus, ethanol synthesized by the ethylene hydration process is mostly used. Ethylene is an essential chemical in the petrochemical industry. Ethylene is traditionally produced through the steam cracking of hydrocarbons, and this method remains the predominant method in the industry. In this process, ethanol is produced by a reversible exothermic reaction between ethylene and water vapor. The process consists of three different steps including reaction, recovery, and purification. Ethylene conversion is about 4–25%, and it is recycled. Ethanol selectivity is 98.5 mol%. Phosphoric (V) acid coated onto a solid silicon dioxide has been used mainly as the catalyst. Phosphoric acid is used as a catalyst, and conversion is 4–25%. Acetaldehyde is produced as a byproduct, which can either be sold or further hydrogenated in order to produce ethanol. The unreacted reactants are separated from the vapor mixture of the reactor in a high-pressure separator and then scrubbed with water to dissolve the ethanol. Ethanol–water mixture forms an azeotrope mixture that needs special distillation techniques, which eventually increase the costs. This process is not economically feasible because ethanol is a cheaper chemical than ethylene. Moreover, it is non-renewable if ethylene is produced by hydrocracking petroleum products. Therefore, validating new technologies that may help to reduce the need for chemicals produced via such kinds of chemical process is necessary.

In this context, this study focused on the development of various kinds of nanostructured ZnO onto Si substrates via chemical route synthesis using both classic solvents and some usual non-toxic beverages to substitute the expensive high purity reagents acquired from specialized providers. To our knowledge, this is the first systematic study involving common beverages as reagents in order to fabricate ZnO coatings onto Si for electronic applications by ACG technique. Moreover, the present study offers comparative information on obtaining nanostructured ZnO coatings with a large variety of bulk and surface morphologies consisting of crystalline nanostructures with high crystallinity with wurtzite type structure suitable for applications that require complex surface structuring with a high number of grain boundaries and large surface-to-volume ratios and individual highly crystalline ZnO nanostructures such as nonlinear optics and some optoelectronic applications. All fabricated materials were thoroughly characterized by SEM, EDX, and XRD, and their PL spectroscopic properties were analyzed.

## 2. Materials and Methods

The general idea of this work was to synthesize ZnO by using chemical reactants and then try to substitute each one of them with nontoxic and mild reagents, using the Aqueous Chemical Growth (ACG) method.

All experiments were performed at relatively low temperatures. During the synthesis of nanostructured ZnO coatings onto Si substrates, the reaction mechanism is the following.
Zn^+2^ + 2OH^−^ ↔ Zn(OH)_2_ ↔ ZnO + H_2_O(R1)

In reaction (R1), the source of zinc cations is typically zinc salts, such as zinc nitrate (Zn(NO_3_)_2_), zinc acetate (Zn(CH_3_COO)_2_), or even metallic Zn. The hydroxyl anions can result from the presence of water, alcohol, and other solvents containing hydroxyl radicals. Moreover, OH^−^ can be formed by the reaction of an amine, such as hexamethylenetetramine (HMTA, (CH_2_)_6_N_4_) or even Ammonium carbonate ((NH_4_)_2_CO_3_), with water.

The pH value variation during the reaction could not be monitored due to the fact that the high-pressure conditions in the autoclaves do not allow this. The effect of pH on this kind of reaction was studied during an older work [17]. It was observed that as the pH values increase, the precipitation of ZnO nanostructures starts earlier compared to lower pH values, but the crystal quality becomes rather poor. The poor crystal quality for large pH values can be attributed to a higher reaction rate, which can be confirmed by the increase in precipitation rate of the material while increasing pH value. From the SEM characterization, it is obvious that the pH values indeed affect the morphology of the as-grown ZnO nanostructures according to each solvent that was used.

First, pure chemical reactants, Zn(NO_3_)_2_ or Zn(CH_3_COO)_2_ and HMTA, were used in different solvents (water, ethanol, etc.). The samples resulting from these experiments are used as reference samples and for direct comparison with others from the existing literature.

Eco-friendly synthesis was achieved by substituting the chemical reactant of HMTA with ammonium carbonate using as a source of baking ammonia sold in the local food market. Baking ammonia has been listed as “Generally Recognized as Safe” (GRAS) by the US Food and Drug Administration (21CFR184.1137) [18]. Baking ammonia is a mixture of ammonium bicarbonate (NH_4_HCO_3_) and ammonium carbamate (NH_2_COONH_4_). It has been used as a primary leavening agent by bakers, before the advent of baking soda and baking powder, because under heating conditions it breaks into carbon dioxide (leavening), ammonia (needs dissipation), and water [19]. The experiments were performed using the same solvents and under the same temperature conditions by using both types of amines.

To our best knowledge, it is the first time that baking ammonia is used for the ACG synthesis of nanostructured ZnO coatings onto Si substrates. Si substrates were chosen because Si technology is still the most important and used in our days for optoelectronic devices. Si was choose to ensure that the proposed method would be compatible with the actual optoelectronic and microelectronic devices technologies and will allow the obtained ZnO materials to be integrated easily with other components in real life applications. Some preliminary studies on the use of baking ammonia to grow photocatalytic ZnO onto PLA 3D printed scaffolds were reported recently by the same research team [20]. After a systematic study of using baking ammonia as the OH^−^ source, the final step for green synthesized ZnO was to substitute the zinc source. According to the literature, ZnO can be obtained by oxidation of metallic Zn powder [21,22,23,24,25]. Consequently, instead of using Zn(NO_3_)_2_ or Zn(CH_3_COO)_2_, Zn metal powder was used. Various experiments were performed by using HMTA and baking ammonia dissolved in various solvents. Several syntheses were also performed without the use of any type of amine.

### 2.1. Materials

ZnO nanostructured coatings were grown by the ACG technique onto well-cleaned silicon (100) wafers obtained from SILCHEM Handelsgesellschaft mbH (Freiberg, Germany) using zinc nitrate hexahydrate Zn(NO_3_)_2_·6H_2_O, ≥99.99%; zinc acetate dihydrate Zn(CH_3_COO)_2_·2H_2_O, ≥99.99%; hexamethylenetetramine (CH_2_)_6_N_4_, >99.00%; and ethylic alcohol, all purchased from, Sigma-Aldrich/Merck KGaA, Darmstadt, Germany and Zn powder from Fluka. Seelze, Switzerland. All of these were used without any further purification. Si wafers were as follows: single side polished; orientation: (100) ± 1°; type: p/boron; resistivity: 15.0 Ohmcm; thickness: 400 ± 25 μm. For the eco-friendly syntheses, nontoxic and daily use reactants were used. Specifically, in addition to water, beverages such as soda water, leavening agents used in traditional puff pastry and cookie recipes such as baking ammonia, mild antiseptics such as hydrogen peroxide (2.8% *w*/*w*), and ethyl alcohol based alcoholic beverages such as Raki and Ouzo (Greek products) were used for the synthesis of the nanostructured ZnO.

### 2.2. Syntheses

The amount of 50 mL of equimolar (0.01 M) aqueous solution of Zn(NO_3_)_2_·6H_2_O and HMTA was placed in a common laboratory oven preheated at a specific temperature (95 °C or 195 °C) for 2 h, followed by washing with deionized water to remove residual contaminants.The amount of 50 mL of equimolar (0.01 M) aqueous solution of Zn(CH_3_COO)_2_·2H_2_O and HMTA was placed in a common laboratory oven preheated at a specific temperature (95 °C or 195 °C) for 2 h, followed by washing with deionized water to remove residual contaminants.Replacing the water with ethanol, raki, and Ouzo for 1 and 2 synthesis conditions.Replacing HMTA with the nontoxic baking ammonia for 1, 2, and 3 synthesis conditions.Replacing the Zn source with Zn metallic. The amount of 1.8 g metallic Zn powder was dissolved in 35 mL of distilled water and 0.05 g HMTA under continuous stirring, followed by thermal treatment in a laboratory oven at 195 °C for 24 h. Washing occurred after.Replacing water in synthesis 5 conditions with ethanol, soda water, lemon soft drink, and hydrogen peroxide (2.8% *w*/*w*).Changing the HMTA amount. Experiments that use 0.15 g instead of 0.05 g HMTA were also performed under the exact same procedure by using the different solvents (water, soda water, lemon soft drink, and hydrogen peroxide).Replacing the HMTA with 0.05 g and 0.15 g nontoxic baking ammonia in synthesis 5 and 6 conditions.Elimination of the amine: 1.8 g metallic Zn powder dissolved in 35 mL of the desired solvent (distilled water, ethanol, soda water, and hydrogen peroxide), followed by thermal treatment in a laboratory oven at 195 °C for 24 h. Washing occurred after.

### 2.3. Characterization

All the prepared samples were characterized by using Scanning Electron Microscopy (SEM) JSM-6390LV SEM from JEOL equipped with Inka-act Energy Dispersive X-ray Analysis (EDX) system from Oxford and X-ray Diffraction (XRD) using using a Rigaku RINT-2000 X-ray diffractometer. XRD patterns were recorded from 30° to 70° 2θ angles using CuKα1 radiation with a monochromatic wavelength of 1.5405 Å operated at 40 kV and 82 mA. Photoluminescence (PL) spectroscopy using a He-Cd, cw laser at 325 nm with full power 35 mW was also performed. The samples were placed in a high vacuum cryostat, which was cooled down to change the temperature from 300 K to 13 K. The emission spectrum was measured using a very sensitive, LN_2_ cooled CCD camera.

## 3. Results and Discussion

### 3.1. Growth and Structuring

Scanning electron microscopy SEM micrographs were analyzed in order to examine the morphology of the prepared samples, while energy dispersive X-ray EDX analysis provided their elemental analysis. Their structural properties were assessed by using XRD measurements via a size-strain Williamson–Hall plot. In addition, the optical properties were studied through PL measurements.

#### Morphology and Structuring onto Si Substrates

SEM images of the samples synthesized via synthesis routes 1, 2, and 3 at 95 °C are presented in Figure 1 for a synthesis temperature of 95 °C.

As it can be observed in Figure 1a, in a water solvent, the use of zinc nitrate and HMTA determined the growth of rod-like nanostructures with high aspect ratios for which the cross section is hexagonal and are randomly distributed onto the substrate. Thus, it seems that the obtained rods preserve the crystal structure of wurtzite ZnO, which is further highlighted from X-ray diffraction analysis. A careful inspection using the SEM analysis software provides an average length and width of ~5 μm and ~500 nm, respectively. Furthermore, the use of zinc acetate and HMTA resulted in flower-like structuring, as one can observe in Figure 1b. Flower-like structures are formed by hexagonal nanorods with a length of ~2–3 μm and a typical width of about 200–300 nm. In addition, several individual rods with ~2–3 μm mean length and larger width of about 300–400 nm can also be observed. It is worth noticing that the use of zinc acetate results in better substrate coverage of ~90%, while zinc nitrate use results in ~80% coverage. The use of ethanol as a solvent results in the formation of co-existing flakes and flower-like structures composed of flakes when using zinc nitrate, as presented in Figure 1c. Flakes thickness ranges between 200 and 800 nm, while their edge length is between 2 and 6 μm, and the substrate coverage is about 80%. By using zinc acetate, we obtained nanospheres with a mean diameter of ~500 nm that reaches a substrate coverage of around 98%. Using ethanol containing Raki traditional Greek drink as solvent results in the dense covering of the Si substrate with flake-like structures in both zinc source cases, as shown in Figure 1e,f. In the case of zinc acetate, the textures are crinklier, and the flakes are denser. Using Ouzo traditional Greek drink as solvent results in a completely different morphology, as presented in Figure 1g,h. For the use of zinc nitrate and HMTA, flower-like structures consisting of nanometer-scale hexagonal rods of typical length and width of ~1 μm and ~500 nm, respectively, were formed. The structuring changes to nanorods with a hexagonal cross section in presence of zinc acetate. The average length of nanorods is ~700–800 nm, while the average width is about 200 nm. The substrate coverage decreased from 80% to 60%, respectively. When the growth temperature becomes 195 °C, the ZnO structuring onto the Si substrate changes dramatically, as can be observed from Figure 2. Figure 2a shows tip rod-like ZnO micro-structures resulting from the aqueous solution of zinc nitrate and HMTA. The tip rod length is ~1.5–3 μm, while the width is ~700 nm. Some bigger rod-like structures are also present, with a length of ~9 μm. In the case of zinc acetate such as a zinc source (Figure 2b), the structuring is flower-like, consisting of tip nanorods with an average length of ~3–5 μm and width of ~600–700 nm. A noticeable increase in substrate coverage from about 70% to 85% can be observed again for zinc acetate use. The use of ethanol results in structuring consisting of the coexistence of tip rods at a of length ~9 μm and width of ~2 μm and flower-like architectures consisting of tip rods with lengths and widths of about 7 μm and ~2 μm, respectively, for zinc nitrate, as shown in Figure 2c. When zinc nitrate is substituted by zinc acetate, the coating consists of spheres, with a mean diameter of ~500 nm (Figure 2d). With the exception of a completely different way of structuring, there is an increase in the coverage of the Si substrate from ~70% to 95%. In the case of Raki use as a solvent, there is a remarkable change in ZnO coating morphology. When zinc nitrate and HMTA are used at 195 °C, ZnO rod-like structures are grown on the Si(100), as observed in Figure 2e. These rods possess a hexagonal cross section, mean length of ~5 μm, and width of ~2 μm. Close morphology is observed when zinc acetate was used, but the rods become uncommon twin hexagonal-truncated-pyramid rod-like structures (Figure 2f). A similar morphology of ZnO was previously observed by Jun Zhang et al. and Suchea et al. [26,27]. The average length and width of the twin hexagonal-truncated-pyramid rod-like structures were ~4–5 μm and ~2 μm, respectively, while substrate coverage increased from ~94% to 98%. Using Ouzo as the solvent, the zinc nitrate-HMTA reaction resulted in spiked mace-like structures composed of rods with a mean length and width of ~2.5 μm and ~250 nm, respectively, as it is observed in Figure 2g. When zinc nitrate was replaced by zinc acetate, nanorods with hexagonal cross section were formed, possessing a mean length of ~600 nm and a width of ~200–300 nm (Figure 2h). The substrates coverage increased again from ~80% to ~85%.

Replacing HMTA with the nontoxic baking ammonia for 1, 2, and 3 synthesis conditions result in a completely different surface morphology of ZnO coatings. SEM characterization images of typical samples obtained at 95 °C are presented in Figure 3.

Figure 3a,c show that replacing HMTA with the nontoxic Baking ammonia for 1 and 2 synthesis conditions result in an obvious decrease in crystallinity and the formation of agglomerated flakes in both cases of using an aqueous solution of zinc nitrate and zinc acetate. The flakes have average widths and edge lengths of about 60 nm and 400 nm, respectively, for zinc nitrate and a mean width of ~100 nm and edge length ~600 nm for zinc acetate use (Figure 3b). The substrate coverage in both cases is ~60–70%. Using ethanol as the solvent resulted in the formation of larger flakes than in the water case with an average width ranging between ~150 and 400 nm and edge length of ~1–2 μm for zinc nitrate, as depicted in Figure 3c, and average width and edge length of about 100 nm and 600 nm, respectively, for zinc acetate use (Figure 3d). In both cases, substrate coverage was ~70–80%. Using Raki as the solvent, the zinc nitrate–baking ammonia reaction resulted in rough surfaces consisting of small agglomerations of about 2 μm diameter consisting of ZnO nanoparticles. When zinc nitrate was replaced by zinc acetate, the morphology remains the same, but the average diameter was ~1–2 μm. The substrates coverages remain ~70% (Figure 3e,f). Using Ouzo as a solvent, the zinc nitrate-baking ammonia reaction resulted in spherical ZnO nanoparticles with a mean diameter size of ~2 μm (Figure 3g), while the zinc acetate–baking ammonia reaction resulted in a more compact coating comprising agglomerated flake structures (Figure 3h). In both cases, the substrates coverages range at ~70–80%.

Performing the same synthesis at a temperature of 195 °C results in the improvement of Si coating crystallinity. SEM images of similar growth conditions as above at 195 °C are presented in Figure 4.

When water was used as a solvent, the presence of zinc nitrate–baking ammonia resulted in quite compact and rough coatings consisting of agglomerations of flakes (Figure 4a). The average width and edge length of flakes are ~100 μm and ~400 nm, respectively. The use of a solution of zinc acetate–baking ammonia in water at higher temperatures resulted in flower-like architectures (Figure 4b) consisting of tip nanorods with a length of several microns (~2–3 μm) and a typical width of around 500 nm. Isolated tip rods with a mean length of ~5–6 μm and width of about 400 nm can also be observed. With the exception of different morphologies, the substrate coverage decreases to 60% in the case of zinc acetate use. By using zinc nitrate and baking ammonia in ethanol, the coatings consist of nonhomogeneous ZnO tip rod flower-like structures (Figure 4c). Some of these structures are composed of dense rods and have a diameter ranging between 2 and 3 μm, while some others consist of more spare rods having a length of ~1 μm and a width of about 300 nm. When zinc nitrate was replaced by zinc acetate, ZnO short rods with hexagonal cross sections were formed (Figure 4d). The average length and width are 1–2 μm and ~300 nm–1 μm, respectively. The substrate coverage remained stable at ~70%. Figure 4e,f, depict the influence of Raki use on the morphology of the samples. More or less spherical agglomerations with rough surfaces and a mean diameter of ~2 μm were observed in both zinc source cases. The Si substrate coverage was ~75–80%. Zinc nitrate–baking ammonia in Ouzo reaction results in typical 1–2 μm diameter sea urchin-like ZnO spherical nanostructures that were formed (Figure 4g). In presence of zinc acetate, agglomerated flakes with typical width and edge length of ~100 nm and 1 μm, respectively, are formed (Figure 4h), resulting in an amorphous dense coating. To summarize, the increased reaction temperatures determine the improvement of coatings morphologies and structure evolution in almost all cases presented above.

Using metallic Zn as a precursor in reaction conditions described in syntheses 5 and 6 determines a complete change of the ZnO growth onto the Si substrates. Changing the HMTA amount from 0.05 g to 0.15 g in synthesis 7 conditions also results in significant growth modifications. Typical SEM characterization images for materials obtained via these syntheses are presented in Figure 5.

Zinc powder with 0.05 g of HMTA dissolved in water resulted in sunflower-like ZnO structures with a diameter ranging between 2 and 5 μm (Figure 5a). As it is shown in Figure 5b, the increase in the amount of HTMA (0.15 g) did not affect the geometry of the structures, while their diameter increased up to ~7 μm. One noticeable difference is that when 0.05 g of HMTA was used, the sunflower-like structures were not well developed and, as a result, their size varies. The increased HMTA amount helps the homogenous growth of structures. The effect of HMTA quantity increase relative to substrate coverage is that it decreases from ~70–80% to a lower percentage. Figure 5c,d illustrate the influence of ethanol in the formation of ZnO structures. Mushroom-like architectures were developed with both amounts of HMTA. Similar morphologies have been previously reported by Jinzhou Yin et al. [28]. Substrate coverage increases from 40 to 60%, with an increasing the amount of HMTA. Using soda water as the solvent results in the formation of tip rods with a mean length of ~8 μm and width ~1 μm, as depicted in Figure 5e. Simultaneously with the presence of tip rods, agglomerated rods with hexagonal cross sections were also formed, as observed from the inset of the figure. The 0.15 g of HMTA determines the formation of hexagonal cross section rod-like structures, and the ~5 μm diameter flower-like agglomerations are formed by the flakes, as observed from Figure 5f. The rods forming the rod-like structures possess lengths of ~6 μm and widths of ~700 nm. The substrate coverage is about the same (80–85%). The use of lemon beverage results in flake structures for both amounts of HMTA (Figure 5g,h). The 0.05 g of HMTA resulting coating flakes show a width of~100 nm and an edge length of ~500 nm, while the respective dimensions when 0.15 g of HMTA was used are ~100 nm and ~600 nm. Finally, hydrogen peroxide (2.8% *w*/*w*) use determines the growth of coatings consisting of a combination of flakes and agglomerated hexagonal rods when 0.05 g of HMTA was used (Figure 5i). The width and edge length of flakes are ~250 nm and ~2 μm, respectively, while the rods have a length of ~1–2 μm and a width of ~800 nm. An increase in the amount of HMTA (0.15 g) resulted in coatings structured as clusters of rods, shown in Figure 5j, with an average diameter of ~5 μm and tip rods with a length of ~6 μm and width of ~2 μm.

By replacing the HMTA with 0.05 g and 0.15 g nontoxic baking ammonia in synthesis 5 and 6 conditions, the eco-friendly chemical synthesis effects on ZnO structuring onto Si substrates were studied. Figure 6 shows typical SEM images of the obtained coatings.

Using an aqueous solution of Zn metallic powder and baking ammonia, sunflower-like ZnO structures with a mean diameter of ~3–5 μm were formed (Figure 6a). The addition of more baking ammonia controls the formation of a mixture of various sized nanorods and irregular flower-like agglomerations, suggesting a change of growth mechanism, as observed in Figure 6b. Nanorods with a hexagonal cross sections possessing a length of ~8 μm and width of ~800–900 nm coexist with flower-like structures composed of flakes possessing a mean diameter of ~5 μm. Figure 6c presents the mushroom-like ZnO structures formed by the reaction of zinc powder with 0.05 g of baking ammonia in ethanol. Smaller mace-like structures with rough surfaces result when the amount of baking ammonia was increased (Figure 6d). The substrate coverage decreased as the amount of baking ammonia increased. Using soda water as the solvent resulted in the formation of rods, with an average length ~4 μm and width of ~600 nm, mixed in a mass of a structure composed of flakes with mean width and edge length of ~100 nm and ~600 nm, respectively (Figure 6e). Figure 6f illustrates the result of increasing the amount of baking ammonia; the rod-like structures with hexagonal cross sections developed while the background material appears to be a mixture of nanoparticles and flakes agglomerations. The substrate coverage increased (~90%). By using the lemon beverage as a solvent, fluffy coating structures consisting of thin flakes were formed for both baking ammonia amounts. The substrate coverage increased from 50% to 80% (Figure 6g,h). Irregular structures consisting of hexagonal rods randomly grown from a ZnO compact agglomeration background deposited onto the substrate were the result of hydrogen peroxide used as a solvent, as shown in Figure 6i,j.

Finally, baking ammonia was eliminated, and syntheses were performed by using the eco-friendly route 9. SEM images at ×2500 and ×5000 magnifications of the ZnO samples, prepared with only Zn metallic powder in various solvents via synthesis 9, are presented in Figure 7.

Sunflower-like structures with irregular shapes of different sizes (Figure 7a,b) resulted from the use of aqueous solution as the solvent. The average diameter of these structures is ~6.5 μm. The substrate coverage can be estimated at about 40–50%. Figure 7c,d reveal only a very small coverage of the substrate area (~30–35%) with inhomogeneous ZnO nanoparticles. Small and larger rods, as well as hexagonal flakes, are formed. Soda water solvent use results in the growth of large flower-like structures with a mean diameter of ~10 μm. The flower-like structures consist of flakes with an average width of ~250 nm and an edge length of ~2 μm (Figure 7e,f). The substrate coverage reaches ~85–90%. The use of hydrogen peroxide (2.8% *w*/*w*) solvent results in coatings consisting of an inhomogeneous mixture of long tip rod structures mixed with spherical agglomerations of densely packed ZnO nanorods, as observed in Figure 7g,h. The tip rods have typical lengths of ~8 μm and a width of 500 nm, while the sphere-like structures are composed of nanorods with a mean length of ~1–2 μm and a width ~500 nm, respectively.

At this point, SEM characterization and micrograph analysis depict the profound relationship between ZnO morphology and the synthesis method, revealing an evolution from rod-like nanostructures to nanospheres or sunflower-like morphology. A further understanding of the microstructure can be attained by using X-ray diffraction, which can provide us with information regarding the interplanar distance, size of the crystalline domains, and the lattice strain of ZnO structuring onto Si substrates, as the commercially available high purity reagents used in typical ACG synthesis are replaced with cheaper nontoxic common ingredients coming from the local market in a trial confirming if nanostructured ZnO coatings suitable for optical and optoelectronic applications can be obtained in a cheap and eco-friendly nontoxic manner. Up to this point, from this study, one can observe that some of the proposed eco-friendly chemical routes are quite promising and further tuning of the reaction conditions may result in high quality nanostructured ZnO coatings onto crystalline Si substrates.

In addition to SEM studies, qualitative composition evaluation of all coatings was performed by EDX analysis. Due to the fact that most of the samples required preparation by Au metallization in order to avoid charging under electron beam exposure, EDX accurate quantitative estimation of elemental composition was not possible. Qualitative EDX analysis and rough stoichiometry estimation proved the presence of ZnO formation in all the fabricated samples. Coatings consisting of well-structured materials showed no presence of any other element except Si, Zn, and O. Coatings consisting of mixed morphology materials obtained via eco-friendly routes also showed the presence of some C contamination in various very low percentages. Some typical EDX spectra are presented in Figure 8.

### 3.2. XRD Characterization

XRD characterization of all synthesized materials was performed.

Materials resulting from synthesis 1 and 2 routes resulted in the formation of ZnO wurtzite phase according to JCPDS card no. 36-1451. In Figure 9a, the X-ray diffraction pattern presents a typical set of diffraction peaks of wurtzite ZnO (card no. 079-0208) that can be assigned to (100), (002), (101), (102), (110), (103), (200), (112), and (201) reflections, as observed in Figure 9. In the cases of synthesis 3 and 4 with the use of ethanol, Raki, and Ouzo as solvents, the presence of Zn(OH)_2_ phase was observed, suggesting that ZnO syntheses were not complete. A slight improvement in crystallinity was observed for syntheses at higher temperatures in all situations. Two typical XRD spectra corresponding to the pure phase ZnO coating obtained from synthesis 2 (a) and to the incompletely formed ZnO phase coating obtained via synthesis 3 in the Ouzo solvent (b) are presented in Figure 9.

Furthermore, once the salt precursors were replaced with metallic Zn powder precursor, the nanostructured ZnO coatings also show the occurrence of pure metallic Zn phases in certain situations. Figure 10 reveals XRD patterns for the samples obtained from zinc powder in different solvents.

First, for the Zn powder and HMTA method, five solvents were used, namely water, ethanol, soda water, lemon beverage, and hydrogen peroxide. In the case of water, all diffraction peaks can be assigned unambiguously as ZnO. On the other hand, in the case of ethanol, unidentified diffraction peaks located at 38.99° and 43.19° can be observed. According to card no. 001-1238, these are consistent with metallic Zn with a = b = 0.26 nm and c = 0.49 nm. Furthermore, in the case of soda water and lemon beverage, only a small diffraction feature located at 38.30° corresponding to Zn(OH)_2_ was detected. Finally, in the hydrogen peroxide solvent, pure ZnO was obtained. For the Zn powder and baking ammonia method, pure ZnO can be observed for water and hydrogen peroxide solvent. On the other side, in soda water and lemon beverage solvent, unreacted Zn(OH)_2_ is observed, while a phase combination between metallic Zn and Zn(OH)_2_ occurred in ethanol. For the last synthesis method, pure ZnO was obtained when water, soda water, and hydrogen peroxide solvents were used. At the same time, the presence of metallic Zn is observed again for ethanol. These results suggest the existence of three stages of ZnO formation: (i) full reaction of Zn(OH) in ZnO; (ii) combination between unreacted metallic Zn/ZnO; (iii) unreacted Zn(OH)_2_/ZnO. For instance, a full reaction took place in the case of using water and hydrogen peroxide as solvents for Zn powder and HMTA and Zn powder and baking ammonia reactions, respectively. One can observe that the use of Zn powder also resulted in pure ZnO in the soda water solvent. In stage (ii), a combination between unreacted metallic Zn/ZnO occurred only for the ethanol solvent. Finally, a combination between unreacted Zn(OH)_2_/ZnO appeared in stage (iii), and it is related to soda water and lemon beverage solvents for Zn powder and HMTA and Zn powder and baking ammonia use. The dependence of crystalline features and synthesis method, as well as the used solvent, is obvious. For a deeper understanding of the obtained ZnO crystallinity state and evolution, the size-strain plot Williamson–Hall method [29] was employed. This plot provides insight into how the residual strain in the crystallite distorts the crystal lattice and, hence, results in the broadening of diffraction peaks. In such cases, Williamson–Hall analysis should be used with Equation (1) by adding the two widths, given by crystallite size, *τ* and the lattice strain *ε*:(1)$$\beta cos\theta =\frac{k\lambda }{\tau }+4\varepsilon sin\theta$$
where k is a shape factor of the crystallites taken as 0.9, and λ is the X-ray wavelength (e.g., 1.54 Å). By examining Equation (1), it is clear that the intercept of the linear fit is related to the mean crystallite size, while the slope determines the strain term. The XRD analyses associated Williamson–Hall plots are shown in Appendix A for each synthesis method as a red line. The values of the intercept and slope, as well as the fitting parameter R^2^ that shows the goodness of fit, are listed in each case. Furthermore, the mean crystallite size and the lattice strain were determined and are tabulated in Table 1.

One can observe that the size of the crystalline domain spans over a wide range of values, from 29.4 nm up to 49.5 nm. At the same time, SEM analysis revealed complex morphologies: The smallest value of the crystalline domains corresponds to flower-like structures, while the largest one corresponds to a combination of flakes and agglomerated hexagonal rods. It is worth mentioning that the ethanol solvent resulted in the smallest mean crystallite size in the case of each synthesis method, which is ascribed to a poorer crystal quality in this case. At the same time, grazing incidence XRD patterns indicated the presence of metallic Zn for this case, ascribed above to stage (ii). On the opposite side, the best crystalline quality given by the use of hydrogen peroxide as a solvent is related to a full reaction of Zn(OH) to ZnO, as XRD analysis shows the presence of pure ZnO onto the substrate. Although the mean crystallite size has a large variation span from one synthesis to another, the lattice strain remains small for all materials (e.g., 0.1–0.2%). In the following, it will be shown that the different ZnO crystalline quality (i.e., different density of structural defects) would determine strong differences in photoluminescence (PL) spectra.

### 3.3. Photoluminescence Studies

As demonstrated above from scanning electron microscopy micrographs and X-ray diffraction analysis via Williamson–Hall plots, a variety of ZnO onto Si substrates sample morphologies was achieved. Such ZnO nanostructures have attracted much attention due to excellent high crystalline quality (mean crystallite size larger than 29 nm in each case, reaching even to 49 nm) and small lattice strain, which could be promising candidates for optical and optoelectronic applications.

The direct valence band of the wurtzite ZnO structure is split into three states, commonly called A (also called the light hole band), B (also called the heavy hole band), and C (also called the crystal-field split band) sub-bands, due to crystal field and spin–orbit interactions [2]. The luminescence properties of the obtained ZnO nanostructures have been intensively studied using photoluminescence spectroscopy. Generally, the typical photoluminescence spectrum of the nanostructured ZnO shows its band edge and exciton luminescence in the UV region. It also presents a green-centered broadband (GL) commonly related to deep-level defects [30]. The first of these bands is usually reported in the literature as NBE (near-band edge excitonic), while the second is known as the DLE band (deep-level emission) [31]. The sizeable binding exciton energy of ZnO (60 meV, nearly three times that of gallium nitride [30]) is responsible for the room temperature luminescence of this material, and it persists at temperatures as high as 700 K according to the literature [32]. Usually, the ratio between the integrated spectral intensity of NBE and DLE bands is used to estimate the contribution of recombination due to defect levels [33]. In ZnO nanostructures, the interaction between these bands becomes more complex because a high surface-volume ratio increases the density of surface defects, which strongly affects the processes of photoluminescence emission.

ZnO UV emission (also called near-band emission, NBE) is located close to its absorption edge (3.37 eV) and is produced by excitonic or band–band recombination. Bound excitons are extrinsic transitions and are related to dopants, native defects, or complexes, which usually create discrete electronic states in the bandgap [34] and, at low temperatures, are the dominant radiative channels. When the ZnO structure has high crystallinity, the UV emission is more intense than the emission in the visible region [35]. Although it is known that the NBE emission peak is composed of several peaks and shoulders of less intensity, in many works, its study is carried out indistinctly, analyzing only the sum of these emissions.

The photoluminescence (PL) of ZnO nanostructures also exhibits a DLE band due to the defect emissions. Several photoluminescence emission centers in the visible region are dependent on the synthesis technique and, hence, on the vacancies, surface defects, and morphology of the ZnO nanostructures. The ZnO visible PL mechanism is still far from being fully understood. The emissions in the region from 3.1 eV down to 1.653 eV are usually referred to as DLE. The deep levels denote the allowed levels inside the bandgap of the semiconductor that produces transitions with energy in the visible range of the spectrum. The band broadness is assumed to come from a superposition of many different deep levels (yellow peak, green peak, and blue peak) that emit simultaneously. Some previous work attributed the green and orange luminescence to extrinsic impurities such as Mo, Cu, Li, or Fe [35]. Various research studies show that undoped ZnO also presents photoluminescence peaks in the visible region and is generally ascribed to the electronic transition from single ionized VO^+^ centers to the valence band edge [36,37].

Typical PL spectra of samples synthesized via synthesis routes 1, 2, and 3 at 95 °C and 195 °C are presented in Figure 11.

From Figure 11, one can clearly observe that even at RT almost all the samples exhibit ZnO UV emission NBE located close to its absorption edge (3.37 eV) and is produced by excitonic or band–band recombination. Bound excitons are extrinsic transitions and are related to dopants, native defects, or complexes, which usually create discrete electronic states in the bandgap. The only sample that had no PL emission is the one synthesized in Raki, which behavior is quite expected based on ZnO structuring onto the Si substrate. At RT, the UV emission is centered at 388 nm for water, 375 nm for ethanol, and 372 nm for Ouzo, which corresponds to energies of ~3.196 eV, ~3.307 eV, and ~3.333 eV, respectively. A better estimation of the energy band gap of ZnO can be given at low temperature PL measurements. At 13 K, the NBE emission is centered at 374 nm for water and 366 nm for ethanol and Ouzo. These wavelengths correspond to ~3.316 eV and ~3.388 eV, respectively, and are closer to the value of ~3.37 eV that is estimated as the NBE emission corresponding to the bound exciton (D^0^X) of ZnO. In both spectra, one can observe a broadband emission from 450 nm to 700 nm, and the photoluminescence (PL) of ZnO nanostructures also exhibits a weak DLE band emission due to the defect emissions related to defects or impurities deriving from the solvents. At RT, the sample prepared in the aqueous solution exhibited the highest PL intensity, which at a lower temperature (13 K) was almost double.

By increasing growth temperature, the ZnO samples’ characteristic PL emission strongly changed for all kinds of solvents. At RT, the PL spectra included both the NBE, as well as the DLE (450 nm–650 nm) emissions; at 13 K, the visible emission considerably diminished. The UV emission at RT is centered at 387 nm for water and raki, 382 nm for ethanol, and 377 nm for Ouzo, and the corresponding energies are 3.204 eV, 3.246 eV, and 3.289 eV, respectively. At 13 K, the NBE peaks are centered at 367 nm for water and Ouzo and at 368 nm for ethanol and raki. The corresponding energies are 3.379 eV (water and Ouzo) and 3.370 eV (ethanol and raki), respectively. It is observed that ZnO synthesized in ethanol at 195 °C exhibits the strongest UV emission, which at 13 K becomes almost 14 times higher and, in the spectrum, an obvious shoulder becomes visible at ~380 nm wavelength near the central peak. This corresponds to ~3.26 eV. This value may be associated with a first-order longitudinal optical (LO) replica (DAP-LO). The shift of the two peaks is near the theoretical LO phonon energy in ZnO (70–75 meV) [38]. The broadening of the NBE peak as the temperature increases (see Figure 11) is attributed to the thermal decomposition and ionization of bound excitons, as well as to the strong coupling of phonons and excitons [38]. The photoluminescence intensity increases as the temperature is lowered to 13 K as a consequence of temperature quenching effects.

Replacing HMTA with nontoxic baking ammonia for 1, 2, and 3 synthesis conditions results in completely different surface morphology of ZnO coatings, and their PL characterization also reveals these changes. Typical PL spectra of samples synthesized via synthesis route 4 at 95 °C and 195 °C are presented in Figure 12.

As it can be observed, in this case only the sample synthesized in aqueous solution exhibited low intensity NBE emission. The emission is centered at 382 nm, corresponding at 3.246 eV; its energy is close enough to the energy gap of ZnO. The samples prepared in ethanol, Raki, and Ouzo did not emit in the UV spectral region. Surprisingly, at 13 K, ZnO nanostructured materials synthesized in water and Raki solvents show clear NBE emission. The UV emission for water is centered at 381 nm and 366 nm for Raki, corresponding to energies of 3.255 eV and 3.388 eV, respectively. When the synthesis temperature was 195 °C, the PL performance of the samples improved, as observed in Figure 12. At RT, the ZnO sample synthesized in ethanol has a strong UV emission, accompanied by a weak visible emission. The UV emission is centered at 387 nm for water and 382 nm for ethanol, corresponding to 3.204 eV and 3.246 eV, respectively. At low temperature (13 K), the UV emission of ZnO samples synthesized in water is centered at 363 nm, 368 nm for ethanol, and 378 nm for Raki, corresponding to energies of 3.416 eV, 3.370 eV, and 3.280 eV, respectively. The best PL performance was exhibited to be synthesized in ethanol, and the emission intensity increased about four times when the PL temperature decreased at 13 K.

Using metallic Zn as a precursor in reaction conditions described in synthesis 5 and 6 determined a complete change of ZnO growth onto the Si substrates, and this reflects as well on the PL characteristic emission of the respective ZnO nanostructured coatings, as shown in Figure 13.

Even at RT, all the samples synthesized in different solvents exhibit clear the characteristic NBE emission of ZnO along with a very weak broad emission in the visible spectral region. At RT, the UV emission is centered at 380 nm for water and ethanol, at 384 nm for soda water, at 379 nm for the lemon beverage, and 385 nm for hydrogen peroxide (2.8% *w*/*w*), corresponding to the energies of 3.263 eV, 3.229 eV, 3.272 eV, and 3.221 eV, respectively. At 13 K, all of the peaks attributed to NBE emission are centered at 367 nm, which corresponds to 3.380 eV. The highest PL intensity, at RT, is exhibited by the sample prepared with hydrogen peroxide (2.8% *w*/*w*), which increased four times at 13 K.

The effects of replacing the HMTA with nontoxic baking ammonia in synthesis 5 and 6 conditions and the eco-friendly chemically synthesized ZnO onto Si substrates PL emission spectra were studied and are presented in Figure 14.

One can easily notice from Figure 14 that all of the prepared samples, independently of the solvent, have the characteristic UV emission of ZnO at both temperatures. Furthermore, at RT, a very weak broad emission at the visible spectral region is observed. At RT, the UV emission is centered at 387 nm for water and hydrogen peroxide (2.8% *w*/*w*), at 379 nm for ethanol, at 384 nm for soda water, and 378 nm for lemon beverage, corresponding to the energies of 3.204 eV, 3.272 eV, 3.229 eV, and 3.280 eV, respectively. At 13 K the characteristic NBE emission peaks are centered at 367 nm for water, ethanol, soda water, and hydrogen peroxide (2.8% *w*/*w*) and 366 nm for the lemon beverage. The corresponding energies are 3.380 eV and 3.390 eV. The highest PL emission intensity at RT can be observed for the sample prepared with water, and its intensity increases 15 times at 13 K.

Finally, after the baking ammonia was eliminated, and syntheses were performed using the eco-friendly route 9; the obtained nanostructured ZnO materials show the PL characteristic emission spectra presented in Figure 15.

All the prepared samples, independent of the solvent used, show the characteristic UV emission of ZnO at both temperatures. At RT, the very weak broad emission in the visible spectral region is observed. At RT, UV emission is centered at 384 nm for water and soda water, at 380 nm for ethanol, and 386 nm for hydrogen peroxide (2.8% *w*/*w*), corresponding to the energies of 3.229 eV, 3.263 eV, and 3.212 eV, respectively. When the temperature decreased at 13 K, all the peaks attributed to exciton recombination are centered at 367 nm, which corresponds to 3.380 eV. The highest PL intensity at RT is shown by the sample prepared with hydrogen peroxide (2.8% *w*/*w*), and it increases 63 times at 13 K.

Using the PL results, an estimation of quantum yield (QY) can be obtained by evaluating the ratio of RT/LT PL intensity integration. This estimation results in the following observations:(1)The samples with Zinc salts, HMTA, or Baking soda (Figure 11 and Figure 12) show a QY of 10–30%, the highest value for the sample prepared in ethanol.(2)The samples with Zinc powder, HMTA, or Baking soda (Figure 13 and Figure 14) results in QY ranging from 4% up to 20%, with the best value for the sample prepared in hydrogen peroxide.(3)The samples synthesized via synthesis route 9 (Figure 15) show QY values ranging from 3% up to 20%, with the better value corresponding to the sample prepared in soda water.

Figure 16 summarizes, in a comparative manner, the previous results regarding the PL performance for all the samples synthesized during this study. Some samples exhibited a very strong PL characteristic NBE emission peak.

As it can be observed, NBE emission at RT position changes from 3.224 eV (light blue line: zinc powder/baking ammonia in water at 195 °C) to 3.303 eV (red line: zinc salts/HMTA in ethanol solution (95 °C)). The FWHM of the NBE emission PL peak provides information about the homogeneous/inhomogeneous broadening of the natural linewidth of the transition, which can be related to crystallinity disorder. First, validation of the line shape could be conducted by using a Voigt function with a different ratio of Lorentzian/Gaussian component. For instance, a Lorentzian is an ideal line shape, and deviation to a Gaussian indicated disorder. A PL spectrum may also have phonon side bands. The inhomogeneity might be caused by band gap fluctuations, which can be due to chemical fluctuations or electrostatic potential variations. The latter can also be related to the degree of compensation, i.e., the ratio of donors to acceptors. If there is a high density of donors or acceptors locally, which cannot be screened, then the band gap fluctuations are observed, resulting in the broadening of the NBE PL peak. Hence, the FWHM of a peak also provides insight into the doping of the investigated material. If the material is strained and/or elongated, it is known that a built-in (or external) (piezo) electric field results in tilting of the energy levels and, thus, in the broadening of PL peaks. The same effect appears whenever some charges are present around, for example, the surface. Moreover, the increase in laser power in the PL experiment can also result in the broadening of the NBE peak since more and more levels are filled; thus, as the samples are formed of many ZnO nanostructures, their collective emission will blend, and the peaks may become broader. The FWHM of the peaks was evaluated to be about 200 meV for all samples, and a Lorentzian fit for the NBE emission peak was used for all samples. From low temperature PL spectra (13 K), the strong emission observed at ~3.38 eV corresponds to free exciton (FX), as well as emission at lower energy which can be assigned to the donor-bound exciton (D-X) and phonon-assisted transitions. The sample synthesized in aqueous solutions of zinc salts and baking ammonia (black line) does not show the FX emission, while the sample prepared with zinc powder in soda water solvent shows a blue shift 50 meV from the FX peaks. The observation of the emission associated with FX, the high PL intensity, and the small FWHM indicates good crystal quality of ZnO nanostructures, even in the samples grown in solvents such as soda water.

The best PL performance was detected on the following.

ZnO synthesized by zinc salts and HMTA at 95 °C in ethanol solution (QY ~30%).

ZnO synthesized by zinc salts and baking ammonia at 195 °C in ethanol solution (QY ~30%).

ZnO synthesized by zinc powder at 195 °C in soda water solution (QY ~20%).

Finally, one can observe that for, the Zn powder and HMTA method, X-ray diffraction analysis showed the best crystal quality for the hydrogen peroxide solution, for which the mean crystallite size **d** is 49.5 nm; the crystal quality then decreased systematically when using soda water, lemon beverage, water, and ethanol. In the case of ethanol, the mean crystallite size reached 33.6 nm, which corresponds to a relative decrease of 32%. At the same time, using hydrogen peroxide and soda water as solvents also resulted in the highest PL intensity at 295 K. Moving forward to the Zn powder and Baking ammonia method, using the water as a solvent resulted in high crystal quality (e.g., **d** = 42 nm) and concomitantly the highest PL intensity. At the opposite side, ethanol solvent resulted in the worst crystal quality (e.g., **d** = 33.8 nm), as well as the smallest PL intensity. For the last synthesis method, namely the zinc powder method, ZnO with the best crystal quality (e.g., **d** = 44.7 nm) was obtained in hydrogen peroxide, and it is related to the highest PL intensity at 295 K. Moreover, ZnO with the worst crystal quality was obtained in ethanol, presenting also the smallest PL intensity. In light of the above information, it seems that the crystal quality that it is related to the PL intensity, together with the microstructure, plays a pivotal role in designing devices with targeted physical properties.

## 4. Conclusions

This study is focused on the development of various kinds of nanostructured ZnO onto Si substrates via chemical route synthesis using both classic solvents, such as raki and Ouzo, as well as of some usual non-toxic beverages, such as soda water, lemon beverage, or hydrogen peroxide. In this manner, the expensive high purity reagents acquired from specialized providers were successfully substituted in ACG synthesis of ZnO. Scanning electron microscopy micrographs reveal the close relationship between ZnO morphology and the synthesis method, revealing an evolution from rod-like nanostructures, to nanospheres or sunflower-like morphology. Since the microstructure determines different optical properties, disclosure of the main structural parameters becomes mandatory. In this sense, the Williamson–Hall method was used to provide a separate description of the effects given by the size and strain in the total broadening of the diffraction peaks. For instance, for the Zn powder and HMTA method, X-ray diffraction showed the best crystal quality for hydrogen peroxide solution for which the mean crystallite size **d** is 49.5 nm; then, the crystal quality decreased systematically when used soda water, lemon beverage, water, and ethanol. In the case of ethanol, the mean crystallite size reached 33.6 nm, which corresponds to a relative decrease of 32%. At the same time, using the hydrogen peroxide and soda water as solvents also resulted in the highest PL intensity at 295 K. Moving forward to the Zn powder and baking ammonia method, using water as solvent resulted in high crystal quality (e.g., **d** = 42 nm) and concomitantly the highest PL intensity. At the opposite side, ethanol solvent resulted in the worst crystal quality (e.g., **d** = 33.8 nm), as well as the smallest PL intensity. For the last synthesis method, namely zinc powder method, ZnO with the best crystal quality (e.g., **d** = 44.7 nm) was obtained in hydrogen peroxide, and it is related to the highest PL intensity at 295 K. Moreover, ZnO with the worst crystal quality was obtained in ethanol, presenting also the smallest PL intensity. Due to large surface-to-volume ratios and individual highly crystalline ZnO nanostructures, PL investigations proved that direct band-gap ZnO growth onto the Si substrates possesses wurtzite type structure, suitable for applications that require complex surface structuring with a high number of grain boundaries, such as nonlinear optics and some optoelectronic applications.

In summary, in the first set of experiments, zinc nitrate and zinc acetate (zinc salts) and HMTA were reacted in water, ethanol, Raki, and Ouzo solvents. Each synthesis was performed at 95 °C and 195 °C for 2 h. The morphology of the nanostructured ZnO onto Si coatings was characterized by diversity, depending on the zinc precursor and the solvent used, as well as the synthesis temperature. Higher temperatures resulted in the formation of ZnO nanostructures with better crystallinity. A typical example is a sample prepared using Raki as the solvent; the flake-like structure was converted to hexagonal section rods when the synthesis temperature increased from 95 °C to 195 °C. Regarding crystallinity, XRD patterns revealed the hexagonal wurtzite phase of ZnO. The PL characterization of the samples synthesized at 95 °C revealed higher emission intensity from the sample synthesized in aqueous solutions, while the increase in the synthesis temperature rendered ethanol solvent the best. This can be correlated to the ZnO nanostructuring onto the Si substrate and the crystalline structure. Furthermore, the HMTA was replaced by baking ammonia. The derived morphologies were different from the ones that resulted when HMTA was used instead of baking ammonia. The structural characterization of the synthesized samples revealed the existence of Zn(OH)_2_ byproducts along with ZnO. The existence of these byproducts was confirmed by PL characterization. Broad and intense PL emission peaks existed on the visible region of the electromagnetic spectrum. Furthermore, according to the PL characterization, only water and ethanol were the solvents that resulted in materials with UV emission at RT. After the substitution of HMTA by baking ammonia, the zinc salts were substituted by zinc powder. SEM images proved that the use of different solvents resulted in different ZnO structuring, while the XRD patterns revealed the hexagonal wurtzite phase of ZnO. All the obtained samples exhibited the UV emission at RT according to the PL characterization, but soda water and hydrogen peroxide used as solvents permitted the growth of materials with the strongest NBE emission. The simple substitution of HMTA by baking ammonia and the use of zinc sources in the same solvents similar to before results in a new kind of ZnO structuring with new properties. Upon changing the solvent and the amount of baking ammonia, different ZnO morphologies were derived. The UV emission peak was present in the PL spectra of the samples at RT, with higher intensities for the nanostructured ZnO synthesized in water. The last set of studied materials included samples prepared by decomposition of zinc powder in water, ethanol, soda water, and hydrogen peroxide (2.8% *w*/*w*). The morphology changes again according to the different solvents, while XRD characterization revealed the hexagonal wurtzite phase of ZnO. All the samples were characterized by PL spectroscopy at room temperature and 13 K. Typical features, such as NBE and DLE bands, were observed, and different contributions of DLE intensity were interpreted as measures of structural defect densities; the UV emission corresponding to the NBE emission of ZnO was observed at RT, but the sample prepared in hydrogen peroxide solvent had the strongest emission. It was proved that a large variety of ZnO nanostructures and individual nanostructures with high crystallinity and excellent optical properties can be obtained by using some very cheap and facile chemical synthesis routes with high reliability. Further optimization of the desired synthesis route can tune the material properties to achieve the necessary quality needed for a specific application.

The wide range of ZnO morphologies induces different sizes of the crystalline domains, and we proved the further relationship relative to the optical properties. In this context, it is clear that the use of different synthesis methods leads to a tuning of main structural parameters that are further related to different optical properties. This approach can help design devices with targeted optical properties using simple chemical methods.

## Figures and Tables

**Figure 1 nanomaterials-11-02490-f001:**
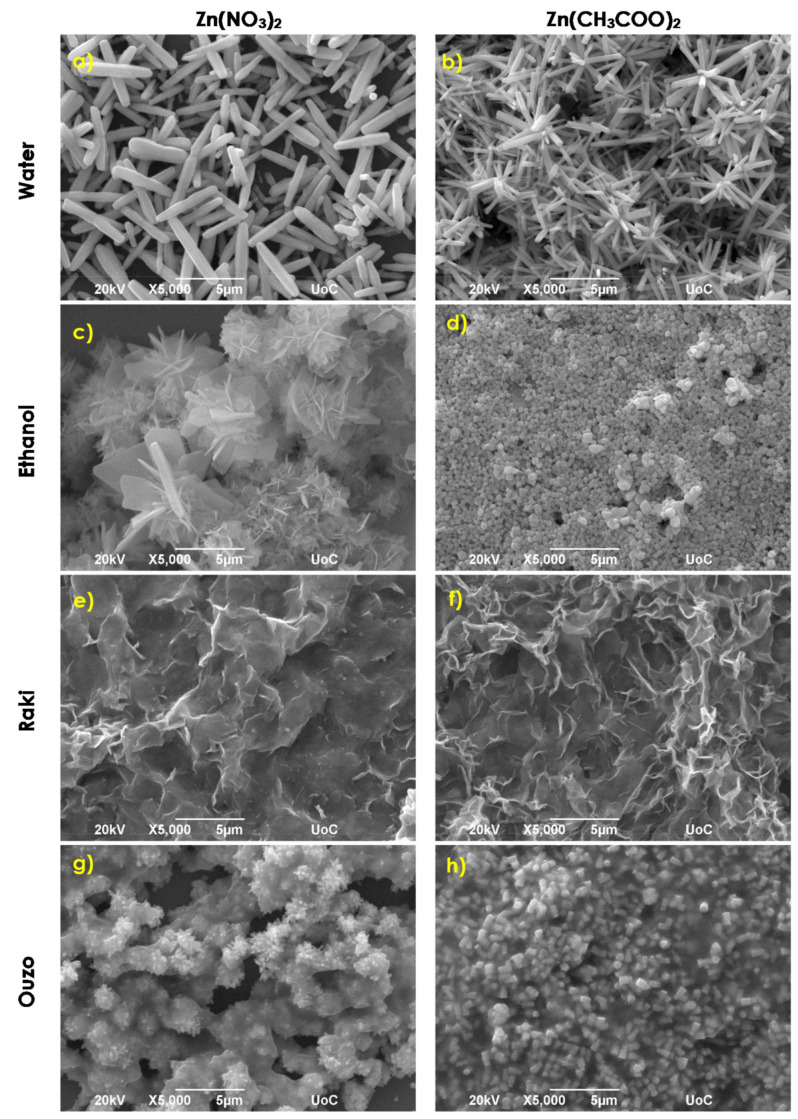
SEM images of ZnO samples synthesized via synthesis routes 1, 2 and 3 at 95 °C.

**Figure 2 nanomaterials-11-02490-f002:**
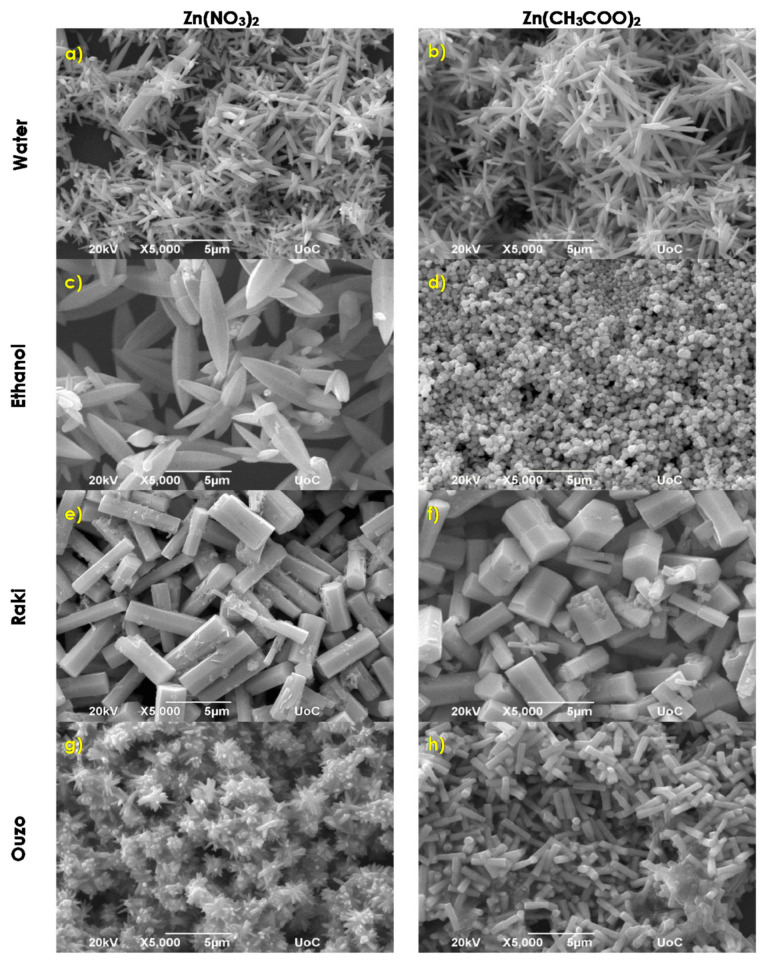
SEM images of ZnO samples synthesized via synthesis routes 1, 2, and 3 at 195 °C.

**Figure 3 nanomaterials-11-02490-f003:**
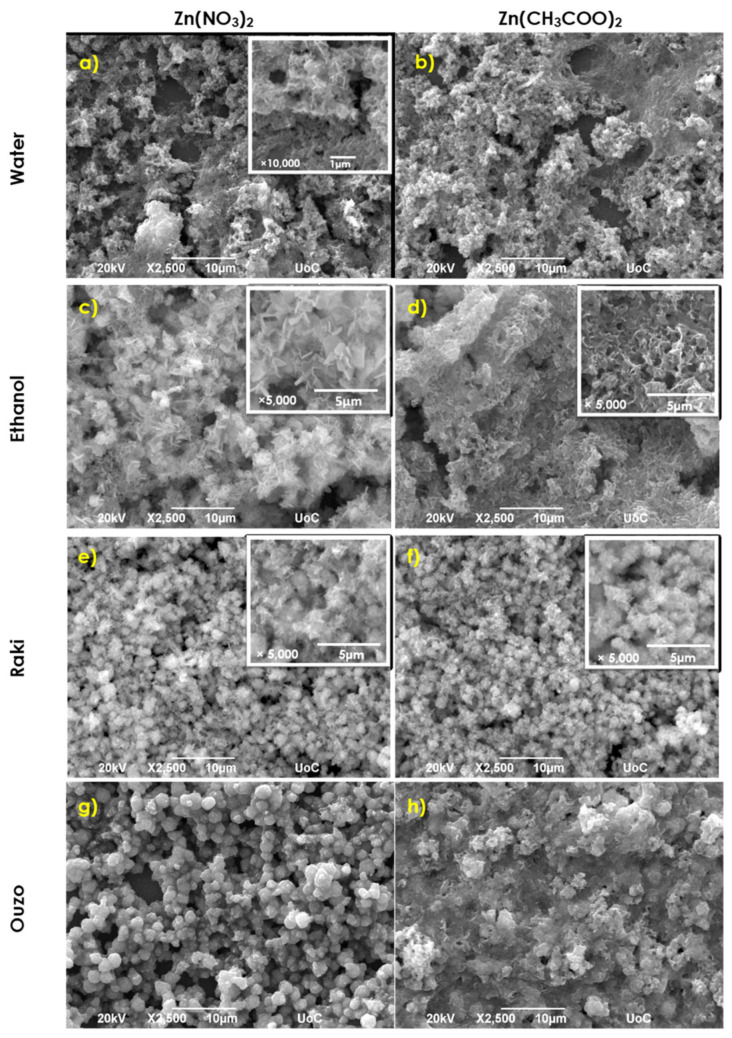
SEM images ×2500 of ZnO samples synthesized via synthesis route 4 at 95 °C. For the case of ZnO nanostructuring zoom images are inserted.

**Figure 4 nanomaterials-11-02490-f004:**
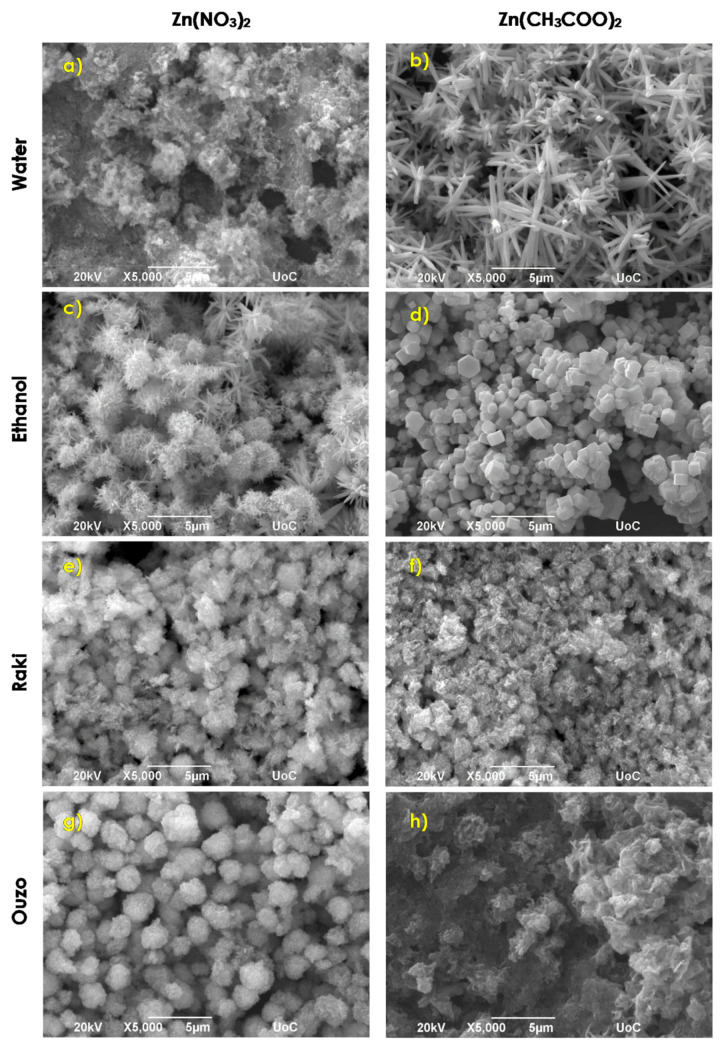
SEM images of ZnO samples synthesized via synthesis route 4 at 195 °C.

**Figure 5 nanomaterials-11-02490-f005:**
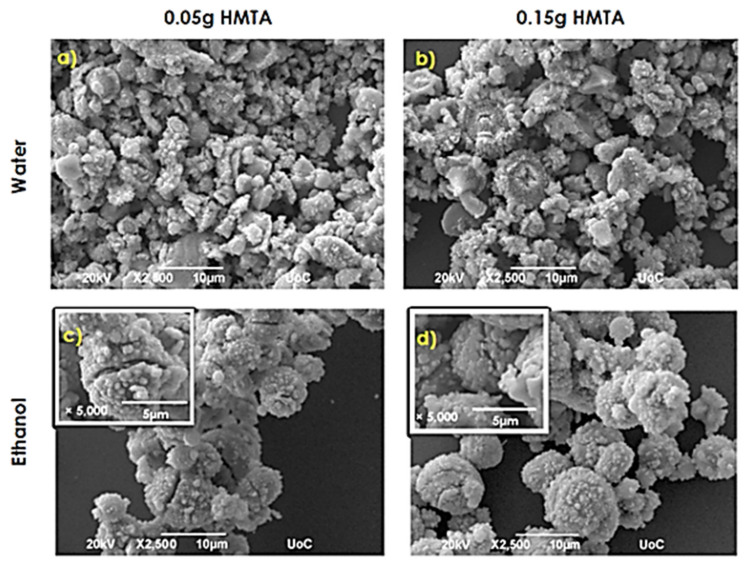
SEM images ×2500 of ZnO samples synthesized via synthesis routes 6 and 7 for samples with smaller features zoom images are inserted.

**Figure 6 nanomaterials-11-02490-f006:**
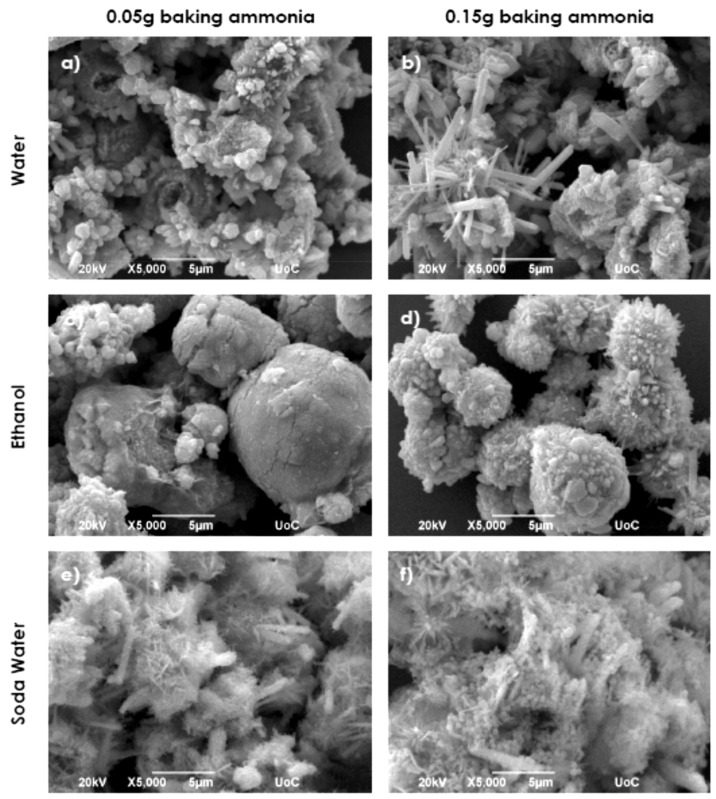
SEM images of ZnO samples synthesized via synthesis route 8.

**Figure 7 nanomaterials-11-02490-f007:**
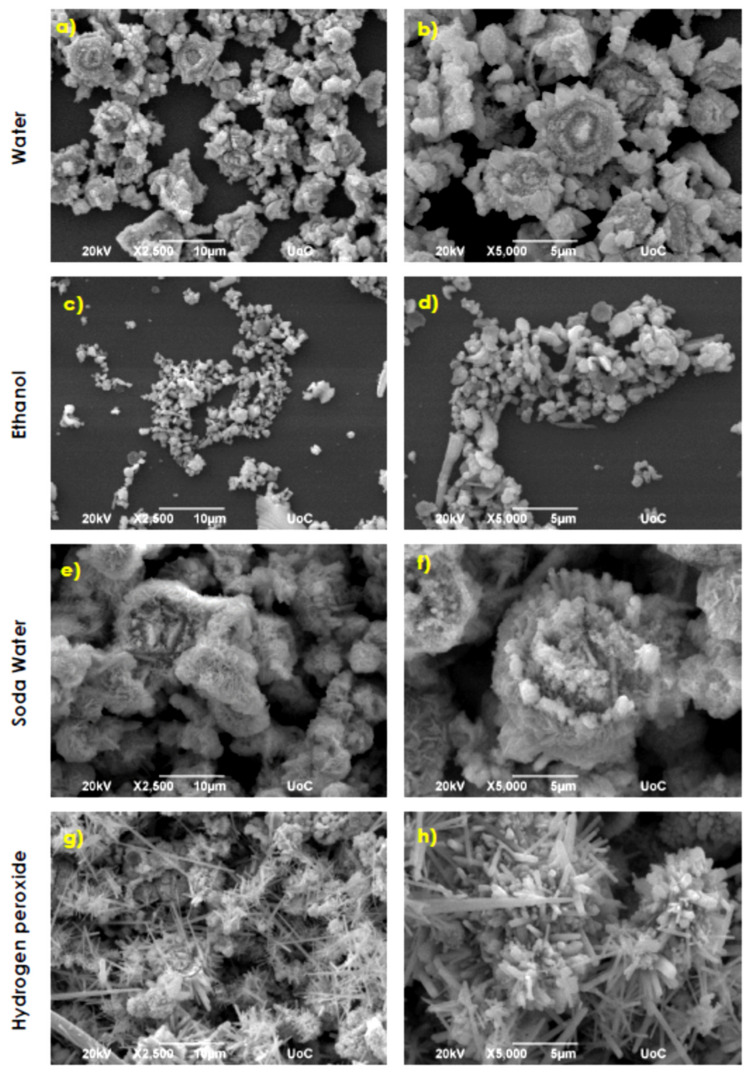
SEM images of ZnO samples synthesized via synthesis route 9 at 195 °C (**a**,**b**) two different magnifications for samples grown in water, (**c**,**d**) two different magnifications for samples grown in ethanol, (**e**,**f**) two different magnifications for samples grown in soda water (**g**,**h**) two different magnifications for samples grown in hydrogen peroxide.

**Figure 8 nanomaterials-11-02490-f008:**
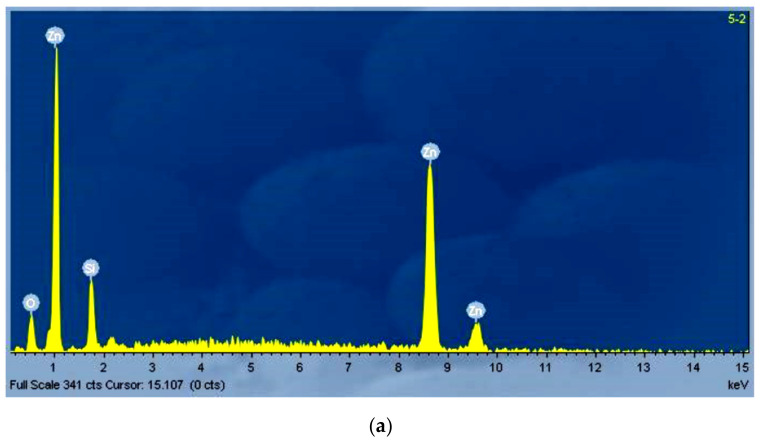
Typical EDX spectra of (**a**) ZnO samples synthesized with zinc nitrate and HMTA at 195 °C in ethanol solvent on Si(100) substrate and (**b**) ZnO samples synthesized with zinc acetate and HMTA at 95 °C in Ouzo solvent on Si(100) substrate.

**Figure 9 nanomaterials-11-02490-f009:**
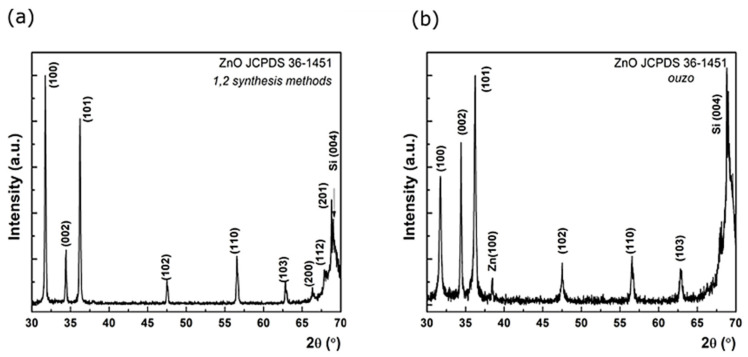
XRD spectra corresponding to the pure phase ZnO coating obtained from synthesis 2 (**a**) and to the incomplete formed ZnO phase coating obtained via synthesis 3 in Ouzo solvent (**b**).

**Figure 10 nanomaterials-11-02490-f010:**
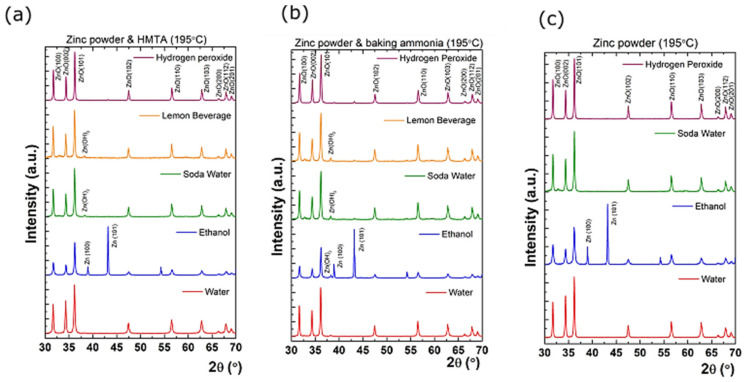
Grazing incidence X-ray diffraction for the synthesized samples from (**a**) Zn powder and Hexamethylenetetramine (HMTA), (**b**) Zn powder and baking ammonia, and (**c**) Zn powder at 195 °C in water, ethanol, soda water, lemon beverage, and hydrogen peroxide, respectively.

**Figure 11 nanomaterials-11-02490-f011:**
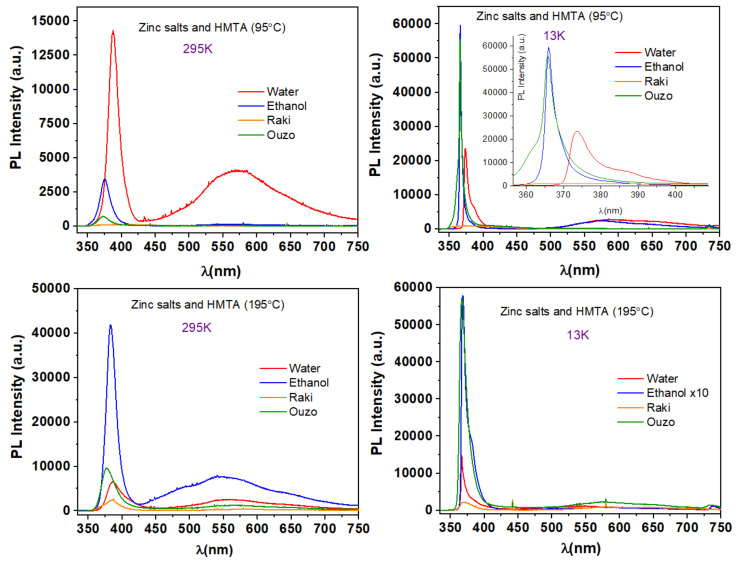
Typical PL spectra of samples synthesized via synthesis routes 1, 2, and 3 at 95 °C and 195 °C.

**Figure 12 nanomaterials-11-02490-f012:**
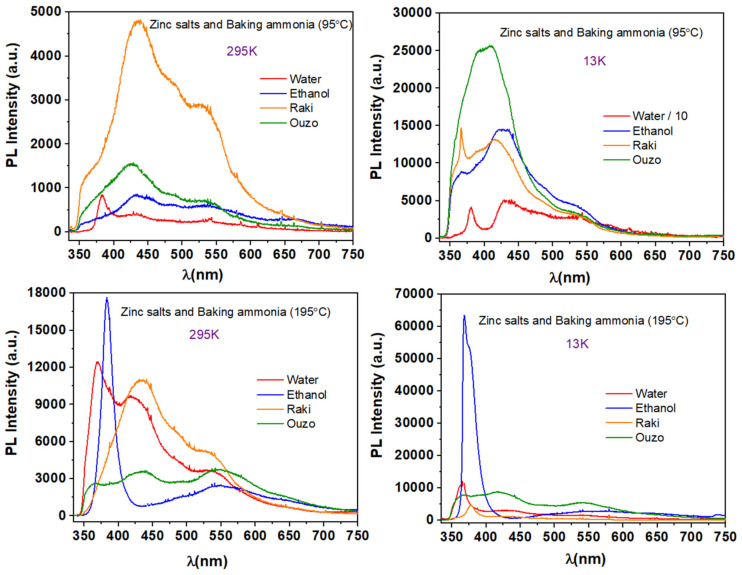
Typical PL spectra of samples synthesized via synthesis route 4 at 95 °C and 195 °C.

**Figure 13 nanomaterials-11-02490-f013:**
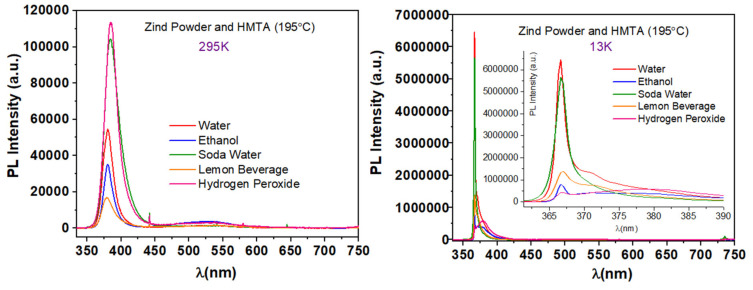
Typical PL spectra of samples synthesized via synthesis route 6 at 195 °C.

**Figure 14 nanomaterials-11-02490-f014:**
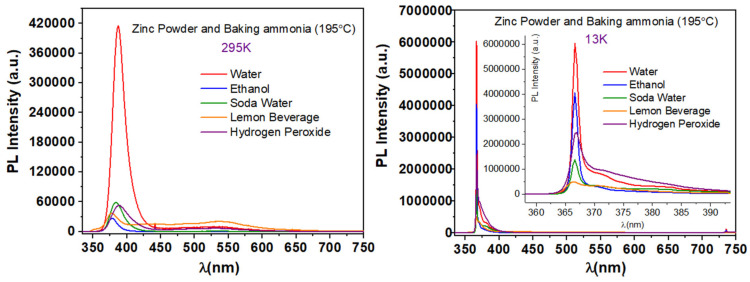
Typical PL spectra of samples synthesized via synthesis route 8 at 195 °C.

**Figure 15 nanomaterials-11-02490-f015:**
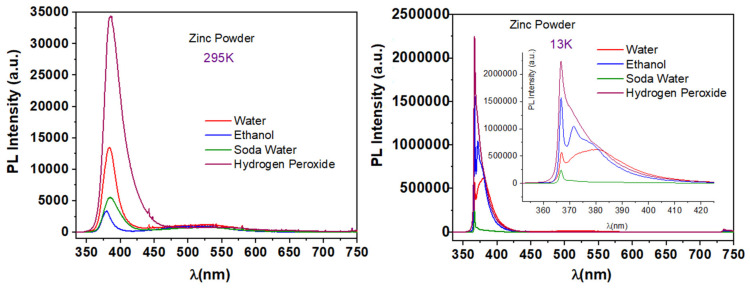
Typical PL spectra of samples synthesized via synthesis route 9 at 195 °C.

**Figure 16 nanomaterials-11-02490-f016:**
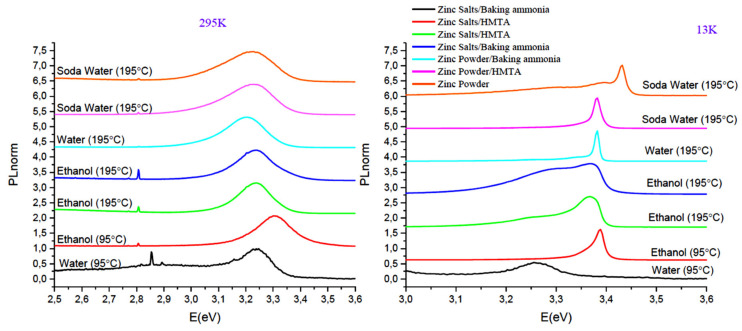
Summary of PL performance for all the samples synthesized during this study.

**Table 1 nanomaterials-11-02490-t001:** Mean crystallite size and the lattice strain derived from Williamson–Hall plot.

Synthesis Method/Solvent	Mean Crystallite Size, d (nm)	Lattice Strain, *ε* (%)
Zn powder and HMTA/water	37.4	0.18
Zn powder and HMTA/ethanol	33.6	0.23
Zn powder and HMTA/soda water	40.7	0.11
Zn powder and HMTA/lemon beverage	40.7	0.13
Zn powder and HMTA/hydrogen peroxide	49.5	0.17
Zn powder and baking ammonia/water	42	0.12
Zn powder and baking ammonia/ethanol	33.8	0.09
Zn powder and baking ammonia/soda water	37.4	0.11
Zn powder and baking ammonia/lemon beverage	36.3	0.11
Zn powder and baking ammonia/hydrogen peroxide	43.3	0.16
Zn powder/water	39.6	0.17
Zn powder/ethanol	29.4	0.15
Zn powder/soda water	40.7	0.14
Zn powder/hydrogen peroxide	44.7	0.12

## Data Availability

The raw and processed data required to reproduce these findings cannot be shared at this time due to technical or time limitations. The raw and processed data will be provided upon reasonable request to anyone interested anytime, until technical problems are be solved.

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
