# Peer review of "Obtaining Nanostructured ZnO onto Si Coatings for Optoelectronic Applications via Eco-Friendly Chemical Preparation Routes"

_nanomaterials, 2021, doi:10.3390/nano11102490_

Round 1

Reviewer 1 Report

The paper presents results on preparation and physical properties of ZnO nanocrystals. The synthesis was done in different solvents (water, ethanol, etc.) and beverages. The obtained results indicate the role of preparation conditions on the morphology and photoluminescence of ZnO nanocrystals. While the paper is interesting and the obtained results are useful for future development of ZnO nanostructures, the following questions can be addressed to the authors to improve the qality of the paper. 1. Besides the HMTA synthesis, it is useful to mention other approaches , e.g. the synthesis in alcohol solutions (see for example J Sol-Gel Sci Technol (2017) 81:333–337. DOI 10.1007/s10971-016-4258-y). 2. The role of Si substrate for the structure and optical properties should be clarified. Otherwise the title is not fully adequate to the content. 3. In order to interpret correctly the PL properties of ZnO nanostructures the information about the PL quantum yield of the investigated samples is also useful.  

Author Response

Thank the reviewer for their constructive suggestions. The manuscript was revised and all the suggestions accommodated.  All the changes in the manuscript are highlighted in blue color font.

Responses to their comments are as follows:

  1. Besides the HMTA synthesis, it is useful to mention other approaches , e.g. the synthesis in alcohol solutions (see for example J Sol-Gel Sci Technol (2017) 81:333–337. DOI 10.1007/s10971-016-4258-y).

“HMTA synthesis” is one of the many chemical ways used for ZnO synthesis, as “the synthesis in alcohol solutions” is too… Although the suggested reference paper is an interesting research on Yttrium doped ZnO preparation and physical properties, it is not really directly related to the present study since the dopant plays an essential role on tailoring material properties and we limit to preparation of undoped ZnO… We preferred to introduce in the manuscript a short comment regarding the growth techniques involved on ZnO thin films in general, and some significant references with respect to growth methods. We hope that the reviewer agrees with our approach.

  1. The role of Si substrate for the structure and optical properties should be clarified. Otherwise the title is not fully adequate to the content.

A paragraph regarding the role of Si and reasons for this choice was introduced in the manuscript. Thank you for your constructive suggestion.

  1. In order to interpret correctly the PL properties of ZnO nanostructures the information about the PL quantum yield of the investigated samples is also useful.  

Usually for an accurate estimation of quantum yield PL measurements with integration sphere and standard reference sample are required. But we couldn’t afford this. However, one can make an estimation of the QY using our kind of measurements from the ration of RT/LT PL intensity integration. The results of the QY estimation by this method were included in the manuscript.

We hope that the reviewer will find the new version of the manuscript suitable for publication.

Reviewer 2 Report

In manuscript entitled “Obtaining nanostructured ZnO onto Si coatings for optoelectronic applications via eco-friendly chemical preparation routes” Authors have described the ZnO on Si synthesis by application of different precursors of zinc ions (Zn(NO3)2 6H2O, f Zn(CH3COO)2 2H2O) using baking ammonia as alkali conditioner. Work is interesting.

This paper should be revised according to the following comments.

  1. In introduction part Authors should additional information regarding the green synthesis methods eg. Materials 2020, 13(19), 4347
  2. Please add more detailed information about Si support
  3. In R&D part please comment the influence of pH or pOH and ionic strength of used conditioner for crystallization effects visualized on TEM results. 
  4. Please, discus molecular mechanism of ZnO NPs formation for each of studied conditions. 
  5. Please, carefully revise the manuscript structure and style according to mdpi requirements.  

In my opinion manuscript required the major revision before final acceptance. 

Author Response

We thank the reviewer for their constructive suggestions. The manuscript was revised and all the suggestions accommodated.  All the changes in the manuscript are highlighted in blue color font. The responses  to all their comments are presented bellow:

1. In introduction part Authors should additional information regarding the green synthesis methods eg. Materials 2020, 13(19), 4347

Thank you for your suggestion. Ref https://www.mdpi.com/1996-1944/13/19/4347 was included in Introduction part.

2. Please add more detailed information about Si support

Thank you for your suggestion. Information regarding the Si substrate was added in “Materials and Methods” section of the manuscript.

3. In R&D part please comment the influence of pH or pOH and ionic strength of used conditioner for crystallization effects visualized on TEM results. 

The comment “Unfortunately, we could not measure pH values during the reaction, since the high pressure conditions into the autoclaves would allow us to do so. We have studied the effect of pH in an older work published in Thin Solid Films 515(24):8764-8767; DOI: 10.1016/j.tsf.2007.03.108, noticing that as the pH values increase the precipitation of ZnO nanostructures start earlier compare to lower pH values, but, however, the crystal quality becomes rather poor. The poor crystal quality for large pH values can be attributed to higher reaction rate, which was verified by the increasing of the precipitation rate of the solutions with increasing pH value. Finally, it is clear from the SEM photographs that the pH values indeed affect the morphology of the as-grown ZnO nanostructures” was included in the respective section of the manuscript. We hope that it fulfils the reviewer request.

4. Please, discus molecular mechanism of ZnO NPs formation for each of studied conditions. 

The ZnO material formation mechanism is already presented in the text. The ZnO formation mechanism is the same for all studied cases.

As we have already stated in our manuscript, “…During the synthesis of nanostructured ZnO coatings onto Si substrates, the reaction mechanism is the following:

Zn+2+ 2OH- ↔ Zn(OH)2 ↔ ZnO+H2O (1)

In reaction (1), the source of zinc cations is typically zinc salts, such as zinc nitrate (Zn(NO3)2), zinc acetate (Zn(CH3COO)2), or even metallic Zn. The hydroxyl anions can result from the presence of water, alcohol and other solvents containing hydroxyl radicals…”

In all cases the reaction mechanism of ZnO remains the same as recorded in (1). The only parameter that was changed was the pOH (pH) of the solutions since different alcohols were used and thus different amounts of OH- anions were present in reaction (1).

In the manuscript it is also stated that: “...Moreover, OH- can be formed by the reaction of an amine, such as hexamethylenetetramine (HMTA, (CH2)6N4), or even Ammonium carbonate ((NH4)2CO3), with water..”

As a result, we have (once more) the same reaction [reaction (1)] for the synthesis of ZnO, while different amounts of OH- anions were present since we have increased the quantity of our amine. We kindly ask the reviewer to check our remarks for comment #3.

5. Please, carefully revise the manuscript structure and style according to mdpi requirements.  

Due to some technical issues, this will be done after the revision with the mdpi editorial team approval and help.

In my opinion manuscript required the major revision before final acceptance. 

We hope that the reviewer finds the present improved revised version suitable for publication.

Round 2

Reviewer 2 Report

Revised manuscript can be accepted for publication.